# Long-Term Greenness Effects of Urban Forests to Reduce PM_10_ Concentration: Does the Impact Benefit the Population Vulnerable to Asthma?

**DOI:** 10.3390/ijerph22020167

**Published:** 2025-01-26

**Authors:** Jinsuk Jeong, Chaewan Kim, Sumin Choi, Hong-Duck Sou, Chan-Ryul Park

**Affiliations:** Livable Urban Forests Research Center, National Institute of Forest Science, 57 Hoegiro, Dongdaemun-gu, Seoul 02455, Republic of Korea; diplo522@korea.kr (J.J.); cwkim7865@korea.kr (C.K.); ciromi@korea.kr (S.C.); hongducksou@korea.kr (H.-D.S.)

**Keywords:** PM_10_, PM_2.5_, greenness, blocking forests, asthma, vulnerable population

## Abstract

This study investigates the effect of urban forests in reducing particulate matter (PM) concentrations and its subsequent impact on the number of asthma care visits. Understanding the mechanisms behind the relationship between the greenness of blocking forests and the reduction in PM is crucial for assessing the associated human health benefits. This study analyzed the influencing factors for reducing long-term PM_10_ concentrations, utilizing the vegetation index and meteorological variables. Results showed that the reduction in PM_10_ began in 2011, five years after the establishment of the blocking forest. The annual mean PM_10_ concentrations decreased significantly, driven primarily by summer wind speed and summer Enhanced Vegetation Index (EVI), explaining approximately 62% of the variation. A decrease in the number of asthma care visits was observed, similar to the trend of PM_10_ reduction in the residential area and the increase in the greenness of the blocking forest. The influx of PM into the city, primarily driven by prevailing northwesterly winds, may have been mitigated by the growing blocking forest, contributing to a reduction in asthma-related medical visits among urban residents. In particular, since the onset of the COVID-19 pandemic in 2020, the increase in the PM_2.5_/PM_10_ ratio in residential areas has become more closely linked to the increase in asthma-related medical visits. It suggests another PM_2.5_ emission source in the residential area. The number of asthma care visits among children (under 11) and the elderly (over 65) exhibited a strong positive correlation with PM_10_ levels and a negative correlation with the Normalized Difference Vegetation Index (NDVI). This suggests a link between air quality improvement from the greenness of blocking forests with their capacity to capture PM and respiratory health outcomes, especially for the vulnerable groups to asthma. These findings highlight the need to manage pollutant sources such as transportation and the heating system in residential areas beyond industrial emissions as the point pollution source. The management policies have to focus on protecting vulnerable populations, such as children and the elderly, by implementing small-sized urban forests to adsorb the PM_2.5_ within the city and establishing blocking forests to prevent PM_10_ near the industrial complex.

## 1. Introduction

Particulate matter (PM) is a major air pollutant that significantly contributes to cardiovascular and respiratory illnesses, posing serious public health challenges globally [1,2]. PM_2.5_ exacerbates respiratory diseases by bypassing the nasal mucosa and penetrating deep into the alveoli, impairing lung function and increasing the risk of chronic conditions. There is an increased prevalence of asthma and lung disease, as well as an elevated risk of premature mortality connected to the long-term health impacts of ozone and smaller PM_2.5_ [3,4]. For example, 5–10 million annual asthma emergency room visits globally in 2015 could be attributable to PM_2.5_, representing 4–9% of global visits [5].

Urban forests offer a wide range of ecosystem services, including air pollution removal and a positive effect on human health [6,7,8,9]. In terms of air pollution, the positive effect of urban forest and street trees has been proven in many studies [10,11,12,13]. Green spaces can affect air quality through three mitigation mechanisms of green spaces, particle deposition, dispersion, and modification [14]. The urban forest can capture dust from the air through dry and wet deposition and by adsorbing PM on the surface and epicuticular wax of leaves [15,16,17]. Precipitation can facilitate the removal of PM from the atmosphere by washing PM deposits off leaf surfaces onto the ground [18].

From a public health perspective, urban trees can reduce harm such as air pollution, ultraviolet radiation, heat exposure, and pollen, while enhancing psychological recovery and physical health outcomes [19,20]. There is a wealth of research showing the health benefits and positive impacts of green space, but the underlying pathways linking green infrastructure to human health are still unclear [21,22]. However, green spaces demonstrate significant effects in reducing airborne particulate matter concentrations, operating through multiple mechanisms and varying according to spatial scale, context, and vegetation characteristics [14].

Greenness indexes such as NDVI, EVI have been used to investigate the relationship between air pollutants, green spaces, and human health [1,23,24,25,26,27]. NDVI indicating forest growth status was the most effective parameter for improving the urban forest PM_2.5_ mitigation ability [24]. Residential green spaces, represented by NDVI and EVI, can partially mitigate the negative effects of long-term exposure to air pollutants related to increased prevalent intermediate–high advanced fibrosis [26]. A study using satellite NDVI and detailed land use classification data revealed that residential greenness was associated with reduced mortality from natural causes, respiratory diseases, and cardiovascular conditions [28]. However, there is little empirical evidence linking these positive health benefits of air pollution reduction by urban vegetation [20,21,22].

The Korean government has tried to mitigate the adverse effects of air pollution by establishing new urban forests. Particularly, the blocking forest has been established near the industrial complex areas to keep the spread of air pollutants to the residential area [29,30,31]. The long-term fine dust reduction effect between 2001 and 2018 in the urban forest of the study site was studied, and it was discovered that the concentration in residential areas decreased after 2009 compared to industrial complex areas [30,32]. Nevertheless, some criticism is that the effects of buffer green space in reducing air pollution and mitigating odorant diffusion are insignificant [33]. To effectively manage blocking forests for air pollution reduction, it is essential to understand the various meteorological and blocking forests’ growth related to the reduction capabilities from the period preceding the establishment of the blocking forest to the present. Furthermore, the effectiveness of urban forests in mitigating fine particulate matter and their associated health impacts is also required.

The objective of this study was to compare the greenness of the blocking forest, meteorological variables, and PM_10_ concentrations over a 20-year period, encompassing pre- and post-establishment phases. Additionally, the number of respiratory disease cases was compared to PM_10_ and PM_2.5_ to investigate whether the forest would have significantly reduced the influx of pollutants from a large industrial complex situated within the city and whether it would have positively impacted the health of the city’s residents.

Specifically, in this study, (1) long-term trend analysis of PM_10_ and the multivariable regression with the blocking forests’ greenness and meteorological variables were conducted, (2) the number of asthma care visits was compared with the trend of PM_10_ and the ratio of PM_2.5_/PM_10_ to investigate the health effect of air pollution reduction, especially focusing on the population vulnerable to asthma.

## 2. Study Area

The Sihwa National Industrial Complex in Siheung City represents one of the three largest small-to-medium-sized industrial complexes in South Korea. It was constructed and developed between 1986 and 2006 on the western tidal flats of the country (Figure 1). Given the proximity of the residential area in the northeast to the industrial complex in the southwest, a new blocking forest was established in 2000 to mitigate the impact of air and odor pollutants emanating from the industrial complex driven by prevailing northwesterly winds (Figure 1d).

The location was a tidal flat with high salinity, which presented challenges for vegetation development. Early in development in 2000, the soil was 10 m in height, and 124,000 trees of 21 distinct species, including black pine, were planted. Between 2006 and 2012, more trees were supplemented, and it is approximately 3.5 km long and 200 m wide, constructed in 2000 [30]. Over time, the forest has undergone a notable transformation, exhibiting an increase in both width and density (Figure 2).

## 3. Materials and Methods

### 3.1. PM_10_, PM_2.5_ Data

The fine dust measurement data were obtained from Air Korea (https://www.airkorea.or.kr, accessed on 1 November 2024), and the final confirmed measurement data from 2001 to 2022 were downloaded and used for analysis. The measuring device is a continuous atmospheric fine dust measurement equipment that uses the weight concentration method to collect and weigh fine dust while considering a certain temperature and humidity before and after measurement. To compare fine dust concentrations in the industrial complex and the residential area in the Sihwa area, data from the Sihwa Industrial Complex point were used, while data from the Jeongwang-dong point were used for the residential area. The Gyeonggi Provincial Health and Environment Research Institute operates both branches, measuring PM_10_ (µg/m^3^) and PM_2.5_ (µg/m^3^). The Jeongwang-dong branch, a residential area, began measuring in 1995, while the Sihwa Industrial Complex branch began measuring in 1998.

PM_2.5_ commenced data collection in 2017, rendering a long-term impact study unfeasible. However, the data were employed to examine the health effects. The ratio of PM_2.5_ to PM_10_ was employed to compare the trend of health effects.

### 3.2. Meteorological Data

Environmental data were retrieved from the Meteorological Data Open Portal (data.kma.go.kr, accessed on 1 November 2024) from 2001 to 2022 and gathered at Gaetgol Ecological Park, 724–32 Janggok-dong, Siheung-si, Gyeonggi-do (latitude: N 37.39154, longitude: E 126.77795). The data used came from the Siheung (565) Automatic Weather System (AWS) branch, which is roughly 6.5 km from the fine dust measurement location. It provides temperature (°C), wind direction (degree), wind speed (m/s), and precipitation (mm) at minute, hour, day, month, and year intervals.

### 3.3. The Greenness Indies: EVI and NDVI from Landsat 5, 8

The greenness of forests can be quantified using vegetation indices, such as the Normalized Difference Vegetation Index (NDVI) or the Enhanced Vegetation Index (EVI). These indices have been employed in numerous studies to monitor biophysical characteristics. NDVI is a quantitative measure of vegetation greenness that is useful for understanding vegetation density and assessing changes in plant health. The NDVI is calculated as the ratio of RED and near-infrared (NIR) values, with a range of NDVI values between 1 and −1. As the density and greenness of the vegetation increase, the NDVI approaches a value of 1.$$NDVI=\frac{NIR-RED}{NIR+RED}$$

EVI was developed to optimize the vegetation signal, particularly in high biomass forests, by separating the canopy background signal [34]. The EVI is calculated using RED, near-infrared (NIR) values, and BLUE. The coefficients C1 and C2 equal to 6 and 7.5, respectively, represent the aerosol resistance term. L represents the canopy background adjustment, which addresses nonlinear, differential NIR, and red radiant transfer through a canopy. The scaling factor of 2.5 can be employed to enhance the sensitivity of EVI relative to NDVI. The EVI range is −1 to 1, with healthy vegetation falling between 0.2 and 0.8.$$EVI=2.5\times \frac{NIR-RED}{NIR+C1\times R-C2\times BLUE+L}$$

Landsat 5 and 8 images were employed to analyze the change in NDVI and EVI of the blocking forests from 2001 to 2023 (Appendix A). The analysis used data from Landsat 5 from 2001 to 2011, Landsat 8 from 2013 to 2022, and the mean NDVI and EVI between June and August for each year. The Level 2 product of Landsat, which has already undergone pre-processing (i.e., surface reflectance (SR)), was employed to identify temporal changes in NDVI and EVI. Images with low quality due to clouds or fog were excluded from the analysis, and the NDVI and EVI value was calculated for each remaining image. This study employed Google Earth Engine, a cloud-based platform, to analyze the change in NDVI and EVI on the cloud platform with access to the Landsat image archives [35,36].

It is imperative to ensure the consistency of surface reflectance between different Landsat sensors when data from multiple sensors are combined [37,38,39]. Regarding sensitivity in discriminating vegetation differences, NDVI is more saturated in areas of high biomass forest. In contrast, EVI is recommended due to its ability to remain sensitive to canopy variations [34,40]. Some studies have also employed multi-temporal EVI to model rice crops or to classify vegetation types [41,42]. While there are certain limitations to the use of NDVI for the analysis of greenness, several studies have employed NDVI to observe vegetation changes over time [43,44,45]. In this study, two indices were used to compare the contribution of forest growth to the reduction of PM_10_.

### 3.4. The Respiratory Diseases Statistics: Asthma

The number of asthma care visits in Siheung City was obtained from the medical utilization information on environmental diseases provided by the National Health Insurance Service. This information includes data on rhinitis, asthma, and atopy, which are representative environmental diseases. Asthma data used in this study are classified as J45 and J46 of the Korean Standard Classification of Diseases and Causes of Death 8th Revision (KCD-8) same as J45 (Asthma) and J46 (Status Asthmaticus) of the International Classification of Diseases 10th Revision (ICD-10). Asthma statistics were used to identify temporal changes in the relationship between PM_10_ concentration and the PM_2.5_/PM_10_ ratio. Additionally, the correlation among the vegetation index of the blocking forests, PM_10_ and the number of asthma care visits was analyzed.

### 3.5. Linear Multiple Regression and Correlation Analysis

To indirectly corroborate the blocking forest’s influence, a linear multiple regression analysis was performed using several independent variables to explain the annual average PM_10_ concentration. The independent variables were Siheung City’s average EVI and NDVI from June to August for 22 years (2001–2022), for each season, average annual temperature, average annual wind speed, and average yearly precipitation (MAM: March, April, and May; JJA: June, July, and August; SON: September, October, and November; DFJ: December, February, and January). Variables with low contribution were removed sequentially using the stepwise approach. All analyses were performed using R (version 4) and R studio (version 4) [46]. Comparisons between measurement points were conducted using *t*-tests.

## 4. Results

### 4.1. Change in Annual PM_10_ Concentrations

The mean annual PM_10_ concentrations decreased in both the industrial complex and the residential area. Prior to 2010, the concentration in the residential area was higher; however, after 2015, the concentration in the industrial complex was higher. Since the measurements began in 2001, PM_10_ concentrations at the two measurement points have steadily decreased. However, the slope of decline in the residential area (1.93) is steeper than that in the industrial complex (1.14) by 2022 (Figure 3). In 2009, there was a reversal in PM_10_ concentrations in an industrial complex and a residential area, coinciding with a period of forest expansion. Generally, the higher the wind speed, the more particles are blown away from the industrial complex, thereby reducing the concentration of fine dust due to the filtering effect of the blocking forest.

### 4.2. The Main Factors Contributing to the Reduction of PM_10_

Table 1 presents the results of a multiple linear regression analysis between the annual average PM_10_ concentration and the environmental factors. The summer EVI (JJA_EVI, *p* < 0.0008) and summer wind speed (JJA_W.S, *p* = 0.0004) exhibited significant explanatory power at the 0.01 level of significance, while the spring temperature (MAM_T, *p* = 0.0160), annual average precipitation (ALL_P, *p* = 0.0498), winter wind speed (DJF_W.S, *p* = 0.2746), and the autumn temperature (SON_T, *p* = 0.0177) were found to be significant at the 0.05 level. The adjusted R-squared value was 0.6292 (*p* = 0.0001).

### 4.3. Association Among Urban Forest Greenness and PM_10_, PM_2.5_ Reduction, Human Health

The time series change in the mean value of June-August NDVI of the blocking forests exhibited a general increase over time (Figure 4). However, the years 2001, 2005, 2008, 2012, and 2022 were excluded from the analysis due to poor image quality due to cloud cover. The increase in NDVI and EVI values reflects vegetation growth since the establishment of the blocking forest, resulting in increased density and tree canopy. The NDVI values gradually increased from 0.4 in the early 2000s to 0.7 and above by 2020. EVI values were lower than NDVI, but the temporal increasing trend was similar. The temporal pattern of PM_10_ concentration in the residential area exhibited an inverse relationship with greenness indices, indicating that as greenness levels increased, PM_10_ concentrations decreased.

The concentration of PM_10_ in the residential area and the industrial complex, along with the number of asthma care visits in the city, has declined (Figure 5). The decline in the number of asthma care visits began in 2012, while the reduction in PM_10_ concentration in the residential area and the industrial complex commenced in 2006. There was a temporal discrepancy between the declining points for PM_10_ concentration and asthma care cases, which coincided with the period (2006–2012) when the density and canopy of the blocking forest increased as more trees were planted.

A time series analysis of the number of asthma care visits and the ratio of PM_2.5_ to PM_10_ from 2017 to 2023 indicates that the trend association is not as clear as it might be due to the impact of the pandemic (Figure 6). However, the temporal pattern of the number of asthma care visits exhibits a similar trend to that of the PM_2.5_/PM_10_ ratio in residential areas as compared to industrial complex. From 2021 onward, there has been a decline in the PM_2.5_/PM_10_ ratio in industrial complex, while the PM_2.5_/PM_10_ ratio in residential areas and the number of asthma care visits has exhibited an upward trend.

It is notable that the majority of asthma care visits occurred in individuals under the age of 11 when considering all asthma care visits from 2006 to 2023 (51.24%) (Table 2). This specific demographic has contributed to the overall temporal trend in asthma care visits. According to the result of the correlation, there is a very strong correlation between asthma care visits and PM_10_ in the residential area, with a significant level of 0.001 (Table 3). In particular, the age groups most vulnerable to asthma, 0–11 and 65 and above, exhibited a stronger correlation with PM_10_ than all age groups, with a correlation coefficient of 0.803. The correlation between asthma care visits and NDVI was also significantly strong, while EVI demonstrated no significant correlation with asthma care visits.

## 5. Discussion

### 5.1. Urban Forest Management Strategy Considering the Long-Term PM_10_ Reduction Factors

This study demonstrates that the long-term growth of the blocking forest has resulted in a reduction of PM_10_ in the residential area. The reduction of PM_10_ is significantly influenced by the wind speed of the summer, and the EVI value in the summer season. Establishing blocking forests represents an effective strategy for managing point pollution sources, such as those from industrial complexes [31]. In the study area, the prevailing wind blows from the northwest-west direction during the winter, which is the period of high PM concentration in Korea (Figure 1d). The blocking forest was implemented to prevent PM generated by the industrial complex from traveling downwind into residential areas. This intervention has resulted in reducing PM_10_ concentrations in residential areas in comparison to industrial complex over the long term. To maintain and enhance the capacity of blocking forests to reduce PM_10_, it is necessary to implement appropriate management strategies in light of the findings on the main factors affecting PM reduction.

The inverse correlation between PM_10_ and PM_2.5_ with NDVI suggests that the forest may have the potential to remove air pollutants [24,47,48]. In this study, from 2001 to 2022, EVI and NDVI have increased, indicating the blocking forests have been denser and healthier. As evidenced by the survey of the blocking forest, the proportion of poorly growing black pine decreased from 26.9% in 2006 to 9.5% in 2017, and in the case of *Pinus thunbergii*, tree height reached from 5 to 6 m in 2006 to 8 to 12 m in 2017 [32]. This growing good condition could increase the EVI steadily, which impacted the PM reduction [49]. However, in this study, only the summer EVI was found to significantly influence the reduction of PM_10_. EVI and NDVI are indices that measure vegetation density and health; however, EVI is less sensitive to atmospheric and environmental conditions such as in urban areas or areas with air pollution [40,50].

Meanwhile, the 2017 survey revealed that the tree species composition in the 236,000 m^2^ area was as follows. The most prevalent tree species were in the order of *Pinus thunbergii* (36.8%), *Acer buergerianum* (6.2%), *Pinus rigida* (4.8%), *Metasequoia glyptostroboides* (4.6%), *Prunus yedoensis* (4.5%), and *Styphnolobium japonicum* (3.5%). The types of shrubs and broadleaf trees that have the ability to capture PM_2.5_ from the air are most effective when leaves have fully developed, while in the leafless season, the conifer and mixed tree types are the most effective [15]. Coniferous species have a higher PM_10_ and PM_2.5_ dust-holding capacity per unit leaf area than broadleaf species, and coarse leaves enhance the PM-holding capacity [16,51]. Therefore, the coniferous tree species in the blocking forest have been able to maintain the capacity to remove air pollutants even in winter.

Managing urban forests to reduce PM is not merely a matter of maintaining healthy and vibrant urban forests; it also entails the consideration of air circulations within these forests to enhance the deposition and dispersion of PM blown from the industrial complex by the strong winds [52,53]. The blocking forest density decreased from 23.1 trees/m^2^ to 9.6 trees/m^2^, and the green space increased from 0.97 m^3^/m^2^ to 2.02 m^3^/m^2^ due to continuous thinning efforts. Reducing tree density can enhance the ability of blocking forests to remove PM from the air, primarily by improving airflow and particle deposition efficiency. This optimal density can increase contact, resulting in greater PM capture on foliage or deposition to the ground by dry deposition [53,54]. When trees are too densely packed, air circulation within the forest canopy is restricted, which impedes the ability of PM to reach leaf surfaces and branches, allowing pollutants to linger within the forest [55].

### 5.2. PM Reduction Effect on the Health of Vulnerable Groups and Implications for Forest Management in Residential Areas

It is important to understand the mechanism between the greenness of the established forests and the PM reduction and the associated effect on human health [50]. This study showed that the quantity of PM entering the inner city under the influence of the prevailing westerly winds may have been diminished by the presence of the long blocking forests, which could contribute to reducing the number of cases of environmental diseases, such as asthma, among urban residents.

A number of studies have demonstrated the beneficial impact of urban forests on air quality. The effects of urban forests on air quality and human health in the United States were found that in highly vegetated areas, trees can improve air quality by as much as 16% [56]. The improvement in air quality, measured as a percentage of air pollution removal by trees, accounts for less than 1% in urban forests [7]. Though the amount of air pollutants removed by urban forests is relatively modest, it leads to positive human health effects such as reducing cases and morbidity of respiratory and cardiovascular disease [7,8,57]. Establishing functional forests in industrial cities has been an important method of preventing residential areas from being affected by air pollutants [29,30,58].

The temporal change in the ratio of PM_2.5_/PM_10_ in the residential area was similar to that of the number of asthma care visits since 2017. Particularly, following 2021, the asthma care visits have increased along with the ratio PM_2.5_/PM_10_ in the residential areas, while the ratio PM_2.5_/PM_10_ in the industrial complex has decreased. This suggests that there are other pollution sources related to PM_2.5_ in residential areas, such as vehicular emissions. While PM_2.5_ is primarily produced by combustion or chemical reactions, PM_10_ is often created by mechanical activity (road dust, construction sites) or natural events (dust storms, pollen) [18]. To reduce these pollutants, the Korean government restricts the emission sources of PM from December to March of the following year when the concentration of ultrafine dust increases due to seasonal causes, it is designated as a fine dust seasonal management system, with fine dust emission control measures intensified. The management contents include restrictions on the operation of vehicles with level 5 emissions, an intensive crackdown on the illegal burning of agricultural waste, and a crackdown on emission sources such as the intensive management of dust emissions from workplaces such as construction sites.

Additionally, the number of asthma care visits for vulnerable groups is correlated with PM_10_ and NDVI. Exposure to PM_2.5_ and PM_10_ is associated with childhood asthma prevalence and increased asthma morbidity [59,60,61]. Particularly, associations were found between air pollution levels, such as traffic-related air pollution, and increased incidence of asthma in children [62,63]. This indicates that the government policy on PM should prioritize the vulnerable populations (ages 0–11 and 65+) and their residential areas affected by high PM concentrations. Air pollution exposures as far back in time as childhood and adolescence increase the risk of poor lung health in adulthood [27].

In light of the observed correlation between asthma care visits and greenness (NDVI), it is recommended that several small forests and street trees be created in residential areas or on streets with limited available land, considering the mechanism of air pollutant movement [16,53,64,65,66]. The accumulation of PM was found to be greater in roadside plants than in urban forest plants [16]. Roadside plants or gardens have different collection abilities depending on factors such as traffic volume, planting location, and leaf direction, and central road plants collected five times more fine dust than roadside plants [67,68]. These small forests within urban areas can address the high PM concentration and the lack of forests that cause spatial mismatch in the supply and demand of urban forests’ air purification [48,65,69].

### 5.3. Limitation and Further Research

The greenness index EVI was identified as a significant variable as the main factor in PM_10_ reduction and NDVI as the correlation factor with asthma care visits, respectively. This indicates that greenness, which can explain each phenomenon well, may vary. Further case studies are needed to show the relationship between various greenness and pollutants or diseases.

The temporal limitation of the PM_2.5_ data presented a challenge in comparing it with the PM_10_ and asthma data. This trend has only been observed in recent years, and thus, it is advisable to monitor future changes. Further study is required to establish the relationship between these variables and to identify the underlying cause-and-effect mechanisms.

Additionally, further investigation is required to ascertain the magnitude of the PM reduction effect and its spatial extent of blocking forests. The asthma data used in this study were available at the city level. To understand the relationship between asthma, PM, and the blocking forest, the specific data within the area affected by PM and blocking forest should have been used in the study. However, it was not possible to collect asthma data at the local, place level. Some studies found that greenness, which was a 300 m buffer zone around the participants’ address, was not associated with asthma attacks or asthma diagnoses [14,27]. The causal pathways between urban trees, air quality, and asthma are very complex, with notable distinctions between natural scientific and epidemiological approaches to understanding these connections [22]. Greenness needs to be studied in depth regarding its various characteristics and its relationship with asthma depending on the scale [14]. In the case of this study, a more reasonable approach would be to examine the relationship between the greenness of the entire city and the disease-related data at the city level.

## 6. Conclusions

The objective of this study was to ascertain the impact of the blocking forest on reducing PM concentration with vegetation indices and meteorological variables and its subsequent influence on the number of asthma care visits. The influx of PM into the city, primarily driven by prevailing north-westerly winds, may have been mitigated by the growth of the blocking forest, contributing to a reduction in asthma care visits among urban residents.

The impact of blocking forests on PM_10_ reduction became evident five years after its establishment. The annual mean PM_10_ concentration is mainly explained by summer EVI and summer wind speed with an explanatory power of approximately 62%. The trend of PM_10_ exhibited an inverse relationship with greenness and a positive relationship with asthma care visits. Additionally, approximately three years after the onset of the pandemic, the ratio of PM_2.5_ to PM_10_ in residential areas has become more closely associated with the number of asthma care visits than in industrial complex. Furthermore, the number of asthma care visits among children (under 11 years of age) and the elderly (over 65 years of age) demonstrated a strong positive correlation with PM_10_ levels and a negative correlation with NDVI.

In the perspective of policy based on these findings, it is imperative that policies prioritize the protection of vulnerable populations (children and the elderly) through the reinforcement of PM reduction strategies and the reduction in air pollutant emissions. The blocking forest near the pollution sources can filter the air pollutants and the small-sized urban forests within the residential areas can absorb PM_2.5_ from the sources of transportation and the heating system.

## Figures and Tables

**Figure 1 ijerph-22-00167-f001:**
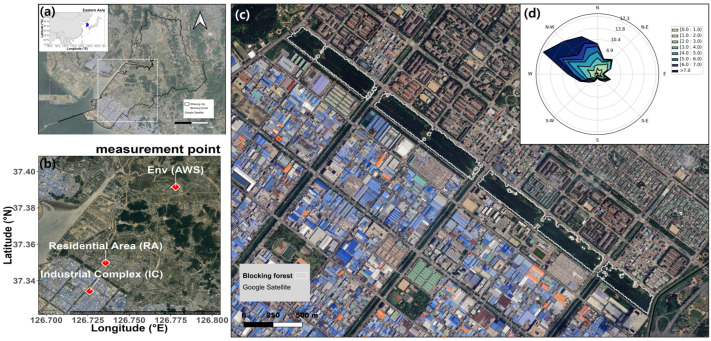
The study area and the PM measurement points (**a**) the location of the study area in South Korea, (**b**) PM measurement points in the study area (IC: Industrial Complex and RA: Residential Area) and the Automatic Weather System (AWS) measurement point for environmental data (Env) located in Siheung City, (**c**) the blocking forest consisting of four-blocked forests, (**d**) the prevailing ground wind direction in the blocking forest during the winter season (December 2018–March 2019) (the higher the value, the higher the wind frequency in that wind direction, and the darker the color, the faster the wind speed).

**Figure 2 ijerph-22-00167-f002:**
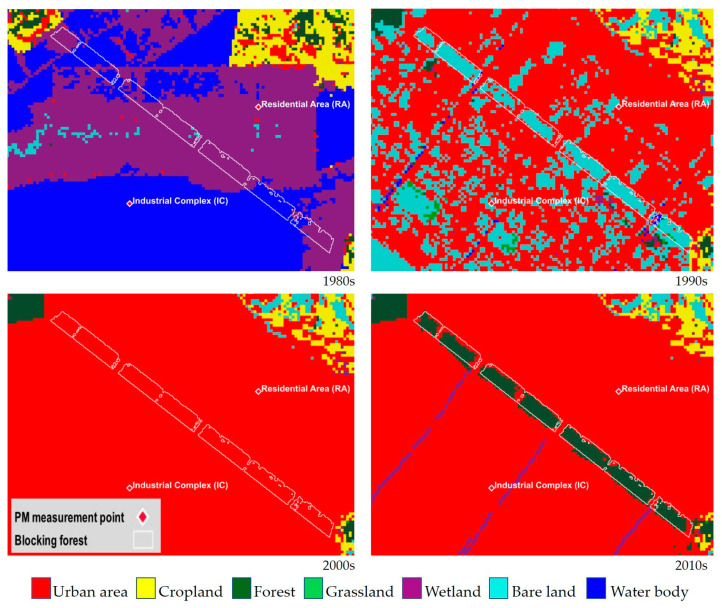
Land cover changes of the blocking forest areas from the 1980s to the 2010s (adapted from the land cover map with 30 m spatial resolution provided by the Korean Environmental Ministry).

**Figure 3 ijerph-22-00167-f003:**
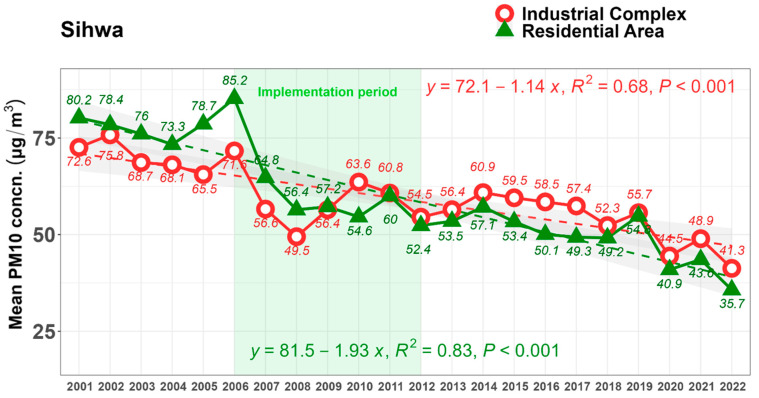
Annual mean PM_10_ concentration trends at two points of the industrial complex and residential area from 2001 to 2022. The implementation of blocking forests started in 2000, and the supplementation of forests was conducted from 2006 to 2012, as illustrated by the light green box.

**Figure 4 ijerph-22-00167-f004:**
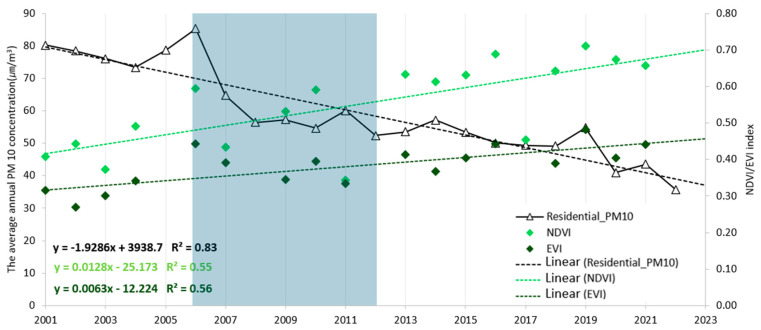
Temporal relationship between observed values of EVI, NDVI, and residentilal_PM_10_ concentration and linear models from 2001 to 2022. (Implementation periods: 2006~2012, shadow area).

**Figure 5 ijerph-22-00167-f005:**
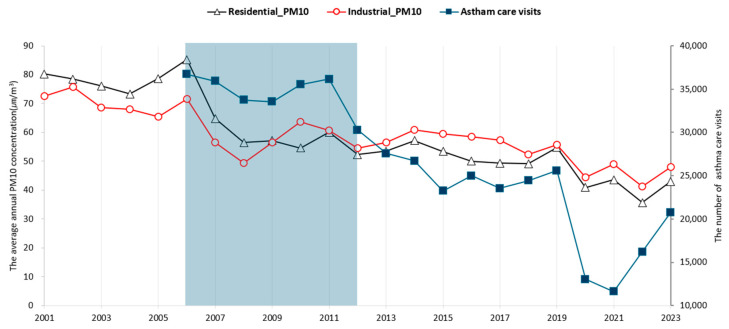
Temporal relationship between PM_10_ and the number of asthma care visits from 2006 to 2023. (Implementation periods: 2006~2012, shadow area).

**Figure 6 ijerph-22-00167-f006:**
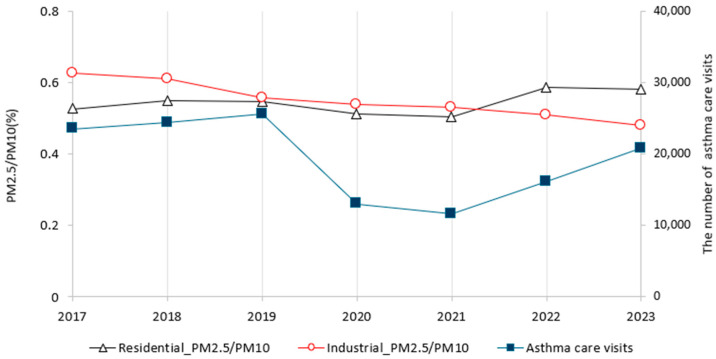
A comparative analysis of the time-series changes in the ratio (PM_2.5_/PM_10_) and the number of asthma care visits from 2017 to 2023.

**Table 1 ijerph-22-00167-t001:** Multiple linear regression estimates and standard errors for the annual mean PM_10_ concentration.

Parameter	Estimate	Standard Error	*p*-Value	Significance
(a) ALL period				
JJA_EVI	−81.4039	21.5256	0.0008	***
ALL_P	−0.0109	0.0053	0.0498	*
DJF_W.S	−28.7971	12.3529	0.0275	*
JJA_W.S	−48.7253	11.9996	0.0004	***
MAM_T	5.7150	2.2229	0.0160	*
MAM_W.S	19.1562	10.5761	0.0812	+
SON_T	4.9698	1.9673	0.0177	*
SON_W.S	37.7498	18.9809	0.0569	+

*p*, significance probability with critical levels set at 0.1 +, 0.05 *, 0.001 ***. All month (ALL), Mar-Apr-May (MAM), Jun-Jul-Aug (JJA), Sept-Oct-Nov (SON), Dec-Jan-Feb (DJF), Precipitation (P), Wind Speed (W.S), Temperature (T), Enhanced Vegetation Index (EVI).

**Table 2 ijerph-22-00167-t002:** Age-group percentage of cumulative asthma care visits from 2006 to 2023.

Age Group	Asthma Care Visits	Percentage (%)
0–5	164,107	34.21
6–11	81,691	17.03
12–17	21,932	4.57
18–44	84,795	17.68
45–64	75,801	15.80
65+	51,404	10.72

**Table 3 ijerph-22-00167-t003:** Correlation among greenness of NDVI and EVI, PM_10_, and age-group-specific asthma care visits.

Correlation Coefficient		NDVI	EVI	PM_10_ (Residential Area)
Asthma care visits	All age groups	−0.586 *	−0.410	0.782 ***
	For vulnerable populations (ages 0–5 and 65+)	−0.593 *	−0.420	0.781 ***
	For vulnerable populations (ages 0–11 and 65+)	−0.597 *	−0.416	0.803 ***

Significance probability with critical levels set at 0.05 *, 0.001 ***.

## Data Availability

Dataset available on request from the authors.

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
