# Peer review of "Long-Term Greenness Effects of Urban Forests to Reduce PM10 Concentration: Does the Impact Benefit the Population Vulnerable to Asthma?"

_ijerph, 2025, doi:10.3390/ijerph22020167_

Round 1

Reviewer 1 Report

Comments and Suggestions for Authors

1) Page9, lines 295-260:  "The reduction of PM10 is significantly influenced by all-season wind speed, particularly during the summer, and the EVI value in the summer season"

This conclusion is not clear in the results introduced in the result section, as the authors did not show the impact of the wind in the results.

2) Page 9, lines 272-273: "However, in this study, only the summer EVI was found to significantly influence the reduction of PM10."

Also, the results and evidence supported this conclusion should be clear.

3) Page 4, beginning of line 135: "1 and -1  As density".

A full stop should be placed before As.

Author Response

Reviewer 1

Suggestions for Authors

Answer: We greatly appreciate your detailed feedback on the manuscript. Thanks to your insightful comments, we could improve the quality of our paper.

1) Page9, lines 295-260:  "The reduction of PM10 is significantly influenced by all-season wind speed, particularly during the summer, and the EVI value in the summer season"

This conclusion is not clear in the results introduced in the result section, as the authors did not show the impact of the wind in the results.

Answer: [Lines 282-284] Thank you for your comments. We changed the expression “The reduction of PM10 is significantly influenced by all-season wind speed, particularly during the summer, and the EVI value in the summer season.” to “The reduction of PM10 is significantly influenced by the wind speed of the summer particularly, and the EVI value in the summer season.

In the result section 4.2, we identified the main factors influencing the PM10, and all-season wind speed has an impact on the PM10 reduction with different significance. But, in this section, we emphasized the wind speed during the summer because the significance is the highest.

2) Page 9, lines 272-273: "However, in this study, only the summer EVI was found to significantly influence the reduction of PM10."

Also, the results and evidence supported this conclusion should be clear.

Answer: We analyzed the relationship between PM10, and the greenness index extracted from the Landsat images to make sure which index is more related. We discussed why EVI has been a significant factor in the reduction of PM10 in section 5.1 (Lines 288-298). But it was not enough to support this result. So, we added about it in section 5.3 ‘Limitation and further research’, suggesting more research is needed because there is limited research to support our result (Lines 323-328).

3) Page 4, beginning of line 135: "1 and -1  As density". A full stop should be placed before As.

Answer: [Line 135] Thank you for correcting the mistake. I corrected it.

Reviewer 2 Report

Comments and Suggestions for Authors

The abstract and the body of work do sync.

The introduction is appropriate and gives the reader a good picture of what the authors are stating in the methodology, results and discussion. Additionally, the references seem appropriate. 

The methods are clearly written and easy to follow. However, the format can be improved by informing the reader on what measurements/ metric SI are used for the different variables such as the meteorological conditions in section 3.2. And are PM2.5 and PM10 (section 3.1) being measured in µg/m³ ? Also can the authors be clear on what ICD codes asthma fall under? this may be very helpful as well. 

The results section is fairly clear and understandable 

The discussion and results are fairly clear. Similarly, you tie the conclusion in well with reasonable limitations.

Overall, it is interesting work but needs some refining and additions in some section as mentioned. Perhaps an additional grammar check.

Author Response

Reviewer 2

The abstract and the body of work do sync.

The introduction is appropriate and gives the reader a good picture of what the authors are stating in the methodology, results and discussion. Additionally, the references seem appropriate.

Answer: Thank you for your thoughtful comments and valuable suggestions.

The methods are clearly written and easy to follow.

However, the format can be improved by informing the reader on what measurements/ metric SI are used for the different variables such as the meteorological conditions in section 3.2.

Answer: We revised this sentence including the unit more concisely and clearly.

[Line 143-145]  “It provides temperature (℃), wind direction (degree), wind speed (m/s), and precipitation (mm) at minute, hour, day, month, and year intervals.”

And are PM2.5 and PM10 (section 3.1) being measured in µg/m³?

Answer: We revised this sentence including the unit more concisely and clearly.

[Line 131-133] “The Gyeonggi Provincial Health and Environment Research Institute operates both branches, measuring PM10 (µg/m³) and PM2.5 (µg/m³).”

Also can the authors be clear on what ICD codes asthma fall under? This may be very helpful as well.

Answer: It is a good suggestion, and so, we added more explanation about the ICD code to the manuscript. Korea uses KCD (Korean Standard Classification of Diseases and Causes of Death), which is essentially made based on ICD, and much of the code is the same as ICD.

[Lines 186-189] “Asthma data used in this study is classified as J45 and J46 of the Korean Standard Classification of Diseases and Causes of Death 8th Revision (KCD-8) same as J45 (Asthma) and J46 (Status Asthmaticus) of the International Classification of Diseases 10th Revision (ICD-10).

The results section is fairly clear and understandable The discussion and results are fairly clear. Similarly, you tie the conclusion in well with reasonable limitations.

Overall, it is interesting work but needs some refining and additions in some section as mentioned. Perhaps an additional grammar check.

Answer: As you suggested, we checked the grammar and refined the overall manuscript. We are grateful for the comprehensive review that has helped enhance the clarity of our work.

Reviewer 3 Report

Comments and Suggestions for Authors

This study explored the impact of blocking forest on reducing PM concentration and asthma visits by determining vegetation indices and meteorological variables, and found that blocking forest can reduce PM10 concentration.

The article seems be well written, easy to follow. The methodology is adequate to achieve the objective. There are couple revisions I would suggest:

1.       In Figure 2, the light green frame indicates the blocking forest, but light green is also used to represent grassland. Might this cause any confusion in interpreting the image?

2.       It would be helpful if the References format could be made consistent throughout.

3.       The Introduction could benefit from providing a more thorough review of the relevant literature in the field.

4.       Are the meteorological data and vegetation indices used adequate to support the validation of the research hypothesis? Have other potential factors that could influence PM concentrations been considered?

5.       Has the sample area been selected in a way that is representative of the broader context? To what extent have potential differences between cities or regions, particularly in terms of climate and geographical factors, been considered?

6.        Does the conclusion perhaps place a relatively stronger emphasis on the role of forest greening in reducing PM concentrations, while there may be other contributing factors, such as policies or industrial emissions, which could also play a significant role?

Author Response

Reviewer 3

This study explored the impact of blocking forest on reducing PM concentration and asthma visits by determining vegetation indices and meteorological variables, and found that blocking forest can reduce PM10 concentration.

The article seems be well written, easy to follow. The methodology is adequate to achieve the objective. There are couple revisions I would suggest:

Answer: Thank you for your thoughtful suggestions and comments that have strengthened our paper. 

  1. In Figure 2, the light green frame indicates the blocking forest, but light green is also used to represent grassland. Might this cause any confusion in interpreting the image?

Answer: [Figures 1 and 2] We changed the color of the boundary of the blocking forest in Fig 1, and 2 to white because as you mentioned, it could be confusing between forests and grasslands.

  1. It would be helpful if the References format could be made consistent throughout.

Answer: We checked the reference format of Int. J. Environ. Res. Public Health and corrected some mistakes in the manuscript.

  1. The Introduction could benefit from providing a more thorough review of the relevant literature in the field.

Answer: First, we deleted some inappropriate literature reviews like [Lines 56-59] Maternal exposure to residential green space was associated with a decreased risk of Preterm birth (PTB, born < 37 completed gestational weeks), with a synergistic effect between low green space and high air pollution levels on PTB [21].“

Instead, to identify the knowledge gap among health benefit(including asthma), greenness index, PM, we added more proper references supporting our research regarding the green spaces' mitigation abilities for air-borne particles from a public health perspective.

[Lines 54-61] “From a public health perspective, urban trees can reduce harm such as air pollution, ultraviolet radiation, heat exposure, and pollen, while enhancing psychological recovery and physical health outcomes [19,20]. There is a wealth of research showing the health benefits and positive impacts of green space, but the underlying pathways linking green infrastructure to human health are still unclear [21,22]. However, green spaces demonstrate significant effects in reducing airborne particulate matter concentrations, operating through multiple mechanisms and varying according to spatial scale, context, and vegetation characteristics [14].

[Lines 67-71] A study using satellite NDVI and detailed land use classification data revealed that residential greenness was associated with reduced mortality from natural causes, respiratory diseases, and cardiovascular conditions [28]. However, there is little empirical evidence linking these positive health benefits of air pollution reduction by urban vegetation [20–22].

  1. Are the meteorological data and vegetation indices used adequate to support the validation of the research hypothesis? Have other potential factors that could influence PM concentrations been considered?

Answer: We greatly appreciate your detailed feedback on the manuscript. There are a variety of factors that have an impact on PM concentrations. Policy and industry change can affect PM concentration, and even vegetation can have different effects on PM depending on the tree species and structures. Depending on the conditions of urban forests and the environment, the effects of PM reduction by urban forests are inconsistent. This study focused on the long-term effects of environmental and vegetation factors. Therefore, we designed our study to be limited to these variables and data to determine the relationship between PM concentrations, vegetation, and asthma.

  1. Has the sample area been selected in a way that is representative of the broader context? To what extent have potential differences between cities or regions, particularly in terms of climate and geographical factors, been considered?

Answer: The PM measurement stations near the industrial complex and residential area bordered by the blocking forests are proper to identify the PM reduction by the forest by comparing the two locations' PM concentration data. However, when we analyzed the association between the health effect (asthma) and PM/blocking forest, PM concentration data collected in one residential area could have a limit to represent the other residential areas of Siheung city. In the next stage, we should do more in-depth research by collecting the other PM measurements and the whole forest within the city or by setting up a study area that is limited to only the buffer area surrounded by the blocking forests with the asthma data from those areas.

So, we described the limitation in section 5.3 limitation and further research carefully. The sentences marked in red are the part that was added to the existing content.

[Lines 381-394]. The asthma data used in this study was available at the city level. To understand the relationship between asthma, PM, and the blocking forest, the specific data within the area affected by PM and blocking forest should have been used in the study. However, it was not possible to collect asthma data at the local, place level. One study found that greenness, which was a 300 m buffer zone around the participants’ address, was not associated with asthma attacks or asthma diagnoses [14,27]. The causal pathways between urban trees, air quality, and asthma are very complex, with notable distinctions between natural scientific and epidemiological approaches to understanding these connections [22]Greenness needs to be studied in depth regarding its various characteristics and its relationship with asthma depending on the scale [14]. In the case of this study, a more reasonable approach would be to examine the relationship between the greenness of the entire city and the disease-related data at the city level.

  1. Does the conclusion perhaps place a relatively stronger emphasis on the role of forest greening in reducing PM concentrations, while there may be other contributing factors, such as policies or industrial emissions, which could also play a significant role?

Answer: We tried to focus on the role of the forest greening policy because this study was about the PM reduction effect of the urban forest. However, we added the government's policy on PM emissions because PM can be reduced more effectively through both PM emission control and PM enhancement by urban forests. We have added the policy on PM emission reduction and reorganized these paragraphs to maintain consistency.

[Lines 343-352]  “While PM2.5 is primarily produced by combustion or chemical reactions, PM10 is often created by mechanical activity (road dust, construction sites) or natural events (dust storms, pollen) [18]. To reduce these pollutants, the Korean government restricts the emission sources of PM from December to March of the following year when the concentration of ultrafine dust increases due to seasonal causes, it is designated as a fine dust seasonal management system, with fine dust emission control measures intensified. The management contents include restrictions on the operation of vehicles with level 5 emissions, an intensive crackdown on the illegal burning of agricultural waste, and a crackdown on emission sources such as the intensive management of dust emissions from workplaces such as construction sites.”

Reviewer 4 Report

Comments and Suggestions for Authors

 The manuscript title “Long-term greenness effects of urban forests to reduce PM10 concentration: Does the impact benefit the population vulnerable to asthma?” presents interesting research on reducing air pollution health impacts. However, several points require improvement:

1. All PM measuring points must be clearly marked on Figures 1 and 2.

2. Figure 3 needs to clearly define and explain the 'implementation period' timeframe.

3. In Table 1, provide full names and definitions for all abbreviated parameters.

4. Mark all asthma care facility locations on Figure 1.

6. Since green barriers are installed in only one direction, include wind rose diagrams or wind direction data to demonstrate whether these barriers effectively block PM from source areas.

7.The study used only 2 PM measuring stations, which is insufficient to conclude that the green barriers can reduce PM concentrations.

Author Response

Reviewer 4

The manuscript title “Long-term greenness effects of urban forests to reduce PM10 concentration: Does the impact benefit the population vulnerable to asthma?” presents interesting research on reducing air pollution health impacts. However, several points require improvement:

Answer: We appreciate your good comments and critical points about our research. We tried to answer to your questions and suggestions to improve and supplement our research.

  1. All PM measuring points must be clearly marked on Figures 1 and 2.

Answer: [Figures 1 and 2] We inserted the PM measurement points in figures 1 and 2.

  1. Figure 3 needs to clearly define and explain the 'implementation period' timeframe.

Answer: As we explained the implementation and supplementation of the blocking forests in the study site section, we added the same brief explanation in Figure 3.

[Line 218-219] “The implementation of blocking forests started in 2000. and the supplementation of forests was conducted from 2006 to 2012, as illustrated by the light green box.”

  1. In Table 1, provide full names and definitions for all abbreviated parameters.

Answer: We added descriptions of all the abbreviated parameters presented in Table 1 at the bottom of the table.

[Lines 231-232]  “All month (ALL), Mar-Apr-May (MAM), Jun-Jul-Aug (JJA), Sept-Oct-Nov (SON), Dec-Jan-Feb (DJF), Precipitation (P), Wind Speed (W.S), Temperature (T), Enhanced Vegetation Index (EVI).”

  1. Mark all asthma care facility locations on Figure 1.

Answer: The asthma data used in this study was available at the city level. In order to understand the relationship between asthma and PM/blocking forest, the specific data within the area affected by PM/blocking forest should have been used in the study. However, it was not possible to collect asthma data at the local, place level. We decided that the limitations of the asthma data needed to be discussed further.

[Lines 382-386] “The asthma data used in this study was available at the city level. To understand the relationship between asthma, PM, and the blocking forest, the specific data within the area affected by PM and blocking forest should have been used in the study. However, it was not possible to collect asthma data at the local, place level.

  1. Since green barriers are installed in only one direction, include wind rose diagrams or wind direction data to demonstrate whether these barriers effectively block PM from source areas.

Answer: Thank you for your insightful comment. The function of blocking forests is closely related to wind direction and should be designed to block the pollutant. To show this relationship, we added more explanation about the prevailing wind direction analysis result in Figure 1.

[Fig 1] “(d) the prevailing wind direction analysis near the study area (the higher the value, the higher the wind frequency in that wind direction, and the darker the color, the faster the wind speed). Additionally, we also added the related description.

[Lines 101-102] “~ driven by prevailing westerly winds (Fig 1(d)).”

  1. The study used only 2 PM measuring stations, which is insufficient to conclude that the green barriers can reduce PM concentrations.

Answer: First of all, we really appreciate your critical point about our study. We agreed with your comment, and this was what we had discussed for a long time. There are six PM monitoring stations in Siheung City, but we only got data from two stations near the industrial complex and the blocking forest. In addition, the asthma data used in this study was available at the city level. Despite the acknowledged limitations of the data set, we pursued this research while carefully considering these limitations in our methodology and interpretation of the results, believing that the findings would still contribute meaningfully to the existing body of knowledge in this area. Thus, we have tried to emphasize this limitation and implication in section "5.3. Limitations and Further Research”.

Round 2

Reviewer 3 Report

Comments and Suggestions for Authors

This paper could be accepted in the current form.

Comments on the Quality of English Language

This paper could be accepted in the current form.

Author Response

Many thanks for your revision and comments on the revised version of ms.

Reviewer 4 Report

Comments and Suggestions for Authors

Please also use a wind rose diagram in the discussion of PM10 reduction in residential areas that are greater than industrial complexes.

Author Response

[comment] Please also use a wind rose diagram to discuss PM10 reduction in residential areas greater than industrial complexes.

[Answer]: [Blue color; Fig 1(d) and Line 115/Discussion 5.1 - Lines 280-281, 284-289, 291]

We replaced the wind rose figure in Fig 1(d) with a more appropriate wind rose analyzed in the blocking forest area during the winter season (2018.12-2019.03), focusing on our study area and the high PM concentration period. Moreover, we further discussed the more PM10 reduction in residential areas than in industrial complexes in discussion section 5.1 with the explanation of the wind rose.